# A Theoretical Approach to Characterize the Accuracy-Fairness Trade-off Pareto Frontier

## Abstract

While the accuracy-fairness trade-off has been frequently observed in the literature of fair machine learning, rigorous theoretical analyses have been scarce. To demystify this long-standing challenge, this work seeks to develop a theoretical framework by characterizing the shape of the accuracy-fairness trade-off Pareto frontier (FairFrontier), determined by a set of all optimal Pareto classifiers that no other classifiers can dominate. Specifically, we first demonstrate the existence of the trade-off in real-world scenarios and then propose four potential categories to characterize the important properties of the accuracy-fairness Pareto frontier. For each category, we identify the necessary conditions that lead to corresponding trade-offs. Experimental results on synthetic data suggest insightful findings of the proposed framework: (1) When sensitive attributes can be fully interpreted by non-sensitive attributes, FairFrontier is mostly continuous. (2) Accuracy can suffer a *sharp* decline when over-pursuing fairness. (3) Eliminate the trade-off via a two-step streamlined approach. The proposed research enables an in-depth understanding of the accuracy-fairness trade-off, pushing current fair machine-learning research to a new frontier.

## 1 Introduction

Fairness has become an essential consideration in algorithmic decision-making, especially in life-critical applications such as healthcare and criminal justice. Unfairness occurs when individuals with higher merit obtain a worse outcome than those with lower merit (Singh et al., 2021). Due to factors such as resource constraints and economic costs, it is often impossible to achieve *complete* fairness and we might have to embrace a certain level of fairness compromise. Hence we establish a set of rules to ensure relatively equitable treatment. For example, the Four-Fifths Rule prescribes that a selection rate for any group (classified by a sensitive attribute) that is less than four-fifths of that for the group with the highest rate constitutes evidence of disparate impact, i.e., discriminatory effects on a protected group. Additionally, prior research has repeatedly observed the tension between fairness and accuracy necessitating complex methods or difficult policy choices (Zhao & Gordon, 2019; Peng et al., 2022). A fundamental question is then: *For a given data distribution, what would the accuracy-fairness trade-off curve look like?*

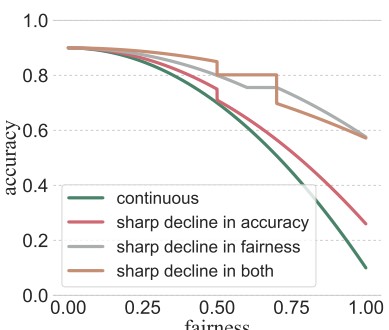

Figure 1: Four shapes of FairFrontier. "Green" delineates a continuous frontier, "Red" exhibits a sharp decline in accuracy when over-pursuing fairness, and "Grey" shows a sharp decline in fairness when improving accuracy. "Brown" represents a sharp decline in both accuracy and fairness.

We refer to the answer as the *accuracy-fairness Pareto frontier (FairFrontier)*. The FairFrontier delineates the optimal performance achievable by a classifier when unlimited data and computing resources are available (Wang et al., 2023). For a fixed data distribution, FairFrontier represents the performance of classifiers that are not dominated by any other classifiers. Characterizing the shape of the FairFrontier enables us to (1) customize the strategy to balance model performance and desired fairness, and (2) evaluate the effectiveness of existing fairness interventions for reduc-

ing algorithmic discrimination. There have been both empirical and theoretical analyses about the accuracy-fairness trade-off(Zafar et al., 2017; Menon & Williamson, 2018). The empirical analysis investigates the trade-off while training a fair machine learning model given a fixed data distribution. It is challenging to control the desired level of fairness since the results highly depend on the datasets and the model architectures. The findings might be problematic due to issues during model training (Cotter et al., 2018), e.g., model miscalibration and sampling bias. By contrast, in theoretical analysis, we can generate synthetic data given a known data distribution and construct optimal classifiers to obtain reliable results and establish fundamental principles for the accuracy-fairness trade-off. Therefore, in this work, a theoretical analysis paradigm is primarily used to mitigate the trade-off (Menon & Williamson, 2018) to conduct the first in-depth examinations of the accuracy-fairness Pareto frontier.

There are many important properties of FairFrontier such as convexity and derivatives. In this work, we focus on one of the fundamental properties: **continuity**. We can categorize the frontier into four types of continuity (Figure 1): continuity, a sharp decline in accuracy, a sharp decline in fairness, and a sharp decline in both fairness and accuracy. This work aims to study under what conditions will these different frontiers occur and then investigate the possibility of eliminating the accuracy-fairness trade-off. Our major contributions are as follows:

- **Characterizing FairFrontier in an Ideal Setting.** We consider an idealized scenario where sensitive attributes can be fully captured by non-sensitive attributes. We show that FairFrontier exhibits continuity in most cases.

- **Characterizing FairFrontier in a Practical Setting.** We further examine the shape of the Fair-Frontier in a more practical setting where non-sensitive attributes encode partial information in the sensitive attributes. We prove that under certain conditions, accuracy may suffer a sharp decline when over-pursing fairness. An upper bound is then derived.

- **Beyond FairFrontier: Eliminating the Trade-off.** We decompose unfairness into data and model unfairness and investigate potential conditions to eliminate the accuracy-fairness trade-off.

## 2 PRELIMINARY

**Notation.** We consider a binary classification task with binary sensitive attributes, with three random variables: the label $Y \in \{0, 1\}$ (the predication label $\hat{Y} \in \{0, 1\}$), the sensitive attribute $A \in \{0, 1\}$, and the Non-sensitive attributes $X$ which satisfies that $X|_{A=a,Y=y} \sim f(x|a, y)$. $\mathbb{P}(\hat{Y} \mid A, Y)$ denotes the probability of the prediction label $\hat{Y}$ given the sensitive attribute $A$ and label $Y$. Let $T_a$ denote the classifier for the sensitive group $A = a$, whereby $\mathbb{P}(\hat{Y} = 1 \mid Y, A = a) = \mathbb{P}(T_a(x) > 0 \mid Y, A = a)$. Similarly, $\mathbb{P}(\hat{Y} = 0 \mid Y, A = a) = \mathbb{P}(T_a(x) < 0 \mid Y, A = a)$, and $T_\theta$ denotes the classifier for the overall distribution.

**Fairness Metric.** Many fairness metrics have recently been proposed (Mehrabi et al., 2022; Zhang et al., 2023). In this paper, we adopt the *Equalized Odds* criterion as the definition of fairness, which requires that the true positive rates (TPR) and the true negative rates (TNR) are equal across all sensitive groups (Hardt et al., 2016). For binary classification tasks, we formally define the true positive rate (TPR) and true negative rate (TNR) with respect to the group $A = a$ as follows:

$$\text{TPR}_{A=a} = \mathbb{P}(\hat{Y} = 1 \mid Y = 1, A = a), \text{TNR}_{A=a} = \mathbb{P}(\hat{Y} = 0 \mid Y = 0, A = a). \quad (1)$$

Hence, we quantify the unfairness $F_U$ under the *Equalized Odds* criterion (Hardt et al., 2016):

$$F_U = \omega_1 \times |\text{TPR}_{A=1} - \text{TPR}_{A=0}| + \omega_2 \times |\text{TNR}_{A=1} - \text{TNR}_{A=0}|. \quad (2)$$

Where $F_U \in [0, 1]$, $\omega_1$ and $\omega_2$ are weights for the TPR and TNR terms, respectively. Considering that the concept of fairness is more familiar, we denote fairness by fairness $= 1 - F_U$, where fairness $\in [0, 1]$. Complete fairness is obtained when fairness $= 1$, namely $F_U = 0$. In this paper, we set the weights $\omega_1 = \omega_2 = \frac{1}{2}$ to equally balance the effects of TPR and TNR in Equation 2. It is noted that other definitions of fairness can also be incorporated into our analysis, and we will leave those to future work. Additionally, we denote the optimal classifier for fairness as $T_a^f$, where

$$T_a^f = \arg\min_{T_a} F_U(T_a). \quad (3)$$

**Accuracy Metric.** For a sensitive group $A = a$, we denote the accuracy of the chosen classifier $T_a$ as $Acc(T_a)$, where

$$
\begin{aligned}
Acc(T_a) =& p_1 \times \mathbb{P}(\hat{Y} = 1, Y = 1) + p_2 \times \mathbb{P}(\hat{Y} = 0, Y = 0) \\
=& p_1 \times \mathbb{P}(T_a(x) > 0 \mid y, A = a) \times \mathbb{P}(Y = 1 \mid A = a) \\
& + p_2 \times \mathbb{P}(T_a(x) < 0 \mid y, A = a) \times \mathbb{P}(Y = 0 \mid A = a).
\end{aligned} \tag{4}
$$

where $Acc \in [0, 1]$, $p_1$ and $p_2$ are weights for the true positive prediction and the true negative prediction, respectively. These parameters underline the desired true predictions. The higher the weight, the more desirable the true prediction would be. In this paper, we set the weighs $p_1 = p_2 = \frac{1}{2}$ to balance the weights of TPR and TNR in Equation 4. Besides, the optimal classifier for accuracy is denoted as $T_a^*$, where:

$$
T_a^* = \arg\max_{T_a} Acc(T_a). \tag{5}
$$

The notation is similar for the overall distribution. Without specification, we refer to the optimal classifier for accuracy as "the optimal classifier".

**FairFrontier**. According to Valdivia et al. (2021), we define a classifier as non-dominated when there is no other classifier that dominates it, i.e., other classifiers cannot improve one objective without worsening the other. Formally in our problem, a classifier $T_\theta$ is said to dominate another classifier $T_\theta^{'}$ if it satisfies one of the following conditions:

$$
\begin{aligned}
\text{Condition 1: } & F_U(T_\theta) \leq F_U(T_\theta^{'}), Acc(T_\theta) > Acc(T_\theta^{'}). \\
\text{Condition 2: } & F_U(T_\theta) < F_U(T_\theta^{'}), Acc(T_\theta) \geq Acc(T_\theta^{'}).
\end{aligned} \tag{6}
$$

According to this definition, a classifier is called a Pareto optimal classifier if there is no other classifier that dominates it, and the set of all optimal Pareto classifiers is defined as the Pareto set. Therefore, our accuracy-fairness trade-off curve, defined as the FairFrontier, can be obtained by measuring the performance of the classifiers of the Pareto set. We can further conclude that the FairFrontier is monotonically non-increasing, starting from the optimal classifier for accuracy and terminating at the classifier that achieves complete fairness (see Figure 1).

## 3  CHARACTERIZING FAIRFRONTIER IN AN IDEAL SETTING

There are many important properties of FairFrontier, such as convexity, derivatives, and so on. This work focuses on the continuity. In this section, we aim to theoretically characterize the shape of the FairFrontier in an ideal setting where sensitive attributes can be fully captured by non-sensitive attributes. Specifically, we will prove that the FairFrontier is mostly continuous and it is impossible that fairness sharply decline or both accuracy and fairness (the grey and brown curves in Figure 1) sharply decline regardless of whether sensitive attributes can be fully encoded by non-sensitive attributes. Full proofs are presented in the Appendix A.

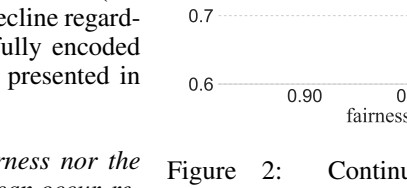

Figure 2: Continuous accuracy-fairness trade-off curve.

**Lemma 1** *Neither the sharp decline in fairness nor the sharp decline in both accuracy and fairness can occur, regardless of whether sensitive attributes are encoded or not.*

Lemma 1 can be obtained by examining the margin of the FairFrontier since the classifier at the point discontinuity has to be the optimal classifier. This lemma indicates that fairness can be continuously and steadily improved by sacrificing accuracy.

**Lemma 2** *When sensitive attributes are fully captured by the non-sensitive attributes, there is a sharp decline in accuracy when over-pursuing fairness* iff. *the point discontinuity represents the local maximum of fairness and the corresponding maximal accuracy with a highly unfavorable prediction for one group.*

*Proof Sketch*: Given the definition of the point discontinuity, any change to the classifier cannot improve fairness, suggesting that fairness has reached the local maximum. Since each point on the trade-off curve corresponds to a classifier on the Pareto frontier, the accuracy at the point discontinuity is the highest for that level of fairness.

Since it is challenging to realize the assumptions in Lemma 2, the major finding in the ideal setting is that the FairFrontier is mostly continuous (the green curve in Figure 1), as we will show later in the Section 6.

# 4 CHARACTERIZING FAIRFRONTIER IN A PRACTICAL SETTING

In reality, non-sensitive attributes are in most cases proxies of sensitive attributes and cannot fully capture the information provided by sensitive attributes. In this section, we investigate how the FairFrontier looks in this practical setting. Specifically, we first investigate whether the accuracy-fairness trade-off exists and then theoretically prove that accuracy sharply declines when the model over-pursues fairness.

## 4.1 THE EXISTENCE OF ACCURACY-FAIRNESS TRADE-OFF

While various empirical findings (Heidari et al., 2018; Friedler et al., 2019; Zafar et al., 2019; Peng et al., 2022) have suggested the existence of the accuracy-fairness trade-off, rigorous theoretical analysis of this observation has been scarce. We aim to complement prior research by considering a simplified setting: We consider a Bayesian optimal classifier and assume the existence of the local maximum of fairness. The assumption is necessary to ensure that the optimally fair classifier is non-trivial: the classifier will not make identical predictions for all samples.

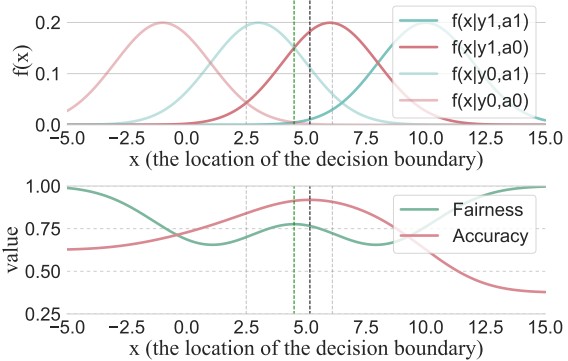

Figure 3: The figure above visualizes the data distributions used in Example 6.1, and the figure below shows the fairness and accuracy when the decision boundary changes. Besides, the green and the black dashed lines correspond to the optimal classifier for fairness and accuracy separately.

We denote the decision boundary of the classifier as $B$, where it can be formulated by $B = \{x | T(x) = 0\}$. We propose the following theorem:

**Theorem 1** *If the classifier $T_\theta$ maximizes both accuracy and fairness simultaneously, then for any sample $x$ on the decision boundary B, it satisfies the following condition:*

$$f(x \mid Y = 1) = f(x \mid Y = 0); \tag{7}$$

*and one of the following:*

*Condition 1:*
$$TPR_{A=0,T_\theta} = TPR_{A=1,T_\theta}, TNR_{A=0,T_\theta} = TNR_{A=1,T_\theta}.$$
*Condition 2:*
$$|f(x \mid Y = 1, A = 0) - f(x \mid Y = 1, A = 1)| = |f(x \mid Y = 0, A = 0) - f(x \mid Y = 0, A = 1)|. \tag{8}$$

*Specifically, if the distributions across different sensitive groups are balanced, i.e. $p(a, y) = \frac{1}{4}$, $\forall x \in B$, and Condition 2 is met, we obtain one of the following conditions:*

*Condition 1:*

$$f(x, A = 1) = f(x, A = 0) = f(x, Y = 1) = f(x, Y = 0).$$

*Condition 2:*

$$f(x \mid A = 0, Y = 0) = f(x \mid Y = 1, A = 0), f(x \mid A = 1, Y = 0) = f(x \mid Y = 1, A = 1).$$
$$(9)$$

This theorem can be derived based on the definition of the optimal classifier. Given the balanced data distributions across different sensitive groups, Condition 1 implies a complete correlation between labels and sensitive attributes (i.e., the correlation coefficient is 1), resulting in a discriminative classifier. Condition 2 stipulates the identical optimal classifiers across different groups, i.e. $T^*_{A=0} = T^*_{A=1}$. However, due to the often impracticability of satisfying these conditions, Theorem 1 suggests that maximizing both accuracy and fairness cannot be achieved simultaneously, i.e., the necessary accuracy-fairness trade-off in reality.

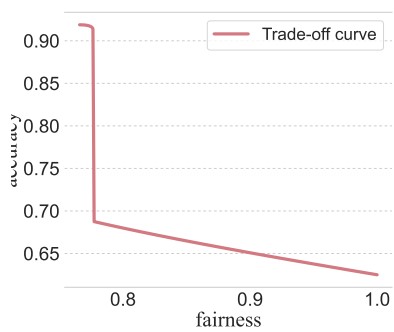

Figure 4: Sharp decline in accuracy when over-pursuing fairness.

## 4.2 THE SHARP DECLINE IN ACCURACY

In Section 3, we identified conditions when the FairFrontier is almost continuous and prove that under no conditions, a sharp decline in fairness alone or in both fairness and accuracy should occur. We will further investigate the conditions that trigger a sharp decline in accuracy (i.e., the red curve in Figure 1).

Given that the sensitive attributes are partially encoded by the non-sensitive attributes, the data distributions of the sensitive groups overlap. We therefore focus on the space enclosed by the decision boundaries of the optimal classifiers for each sensitive group. We further assume that a classifier is well-defined, which states that outside the enclosed space, the sign $(+/-)$ of the predictions is the same as that of the optimal classifier used for each group. This assumption ensures that the classifier can always yield better performance compared to those without this assumption. Formally, a well-defined classifier can be defined as follows:

**Definition 1 (A Well-defined Classifier)** *The classifier $T(x)$ is well-defined if: $\forall x \notin S, T_{A=1}(x) \times T(x) \geq 0$, $T_{A=0}(x) \times T(x) \geq 0$, where $S = \{T_{A=0}(x) \times T_{A=1}(x) \leq 0\}$.*

We then propose the following theorem:

**Theorem 2** *Assuming that the optimal classifier for fairness $T^f_\theta$ is both non-trivial and well-defined, and for any well-defined classifier $T_\theta$, the relationship between $(TPR_{A=1} - TPR_{A=0})(TNR_{A=1} - TNR_{A=0})$ and 0 remains consistent, then the following inequality holds true for the classifier $T_\theta$ that over-pursues fairness:*

$$Acc(T_\theta) \leq \min\{Acc(T^f_\theta), \ \max\{Acc(T^*_{A=0}), Acc(T^*_{A=1})\}\}. \tag{10}$$

*where $Acc(T^*_{A=a})$ represents the accuracy achieved when deploying the optimal classifier for group $A = a$ for both groups. Specifically, under the condition that $\max\{Acc(T^*_{A=0}), Acc(T^*_{A=1})\} < Acc(T^f_\theta)$, the following result holds true:*

$$Acc(T_\theta) \leq \max\{Acc(T^*_{A=0}), Acc(T^*_{A=1})\}. \tag{11}$$

An expected result of Theorem 2 is that over-pursuing fairness can lead to a sharp decline in accuracy. We derive this result by establishing the upper bound that is determined by the optimal

classifier for each sensitive group. If the performance of the optimal classifier for each sensitive group is inferior to that of the fairness-optimized classifier, a sharp accuracy decline becomes inevitable. In addition, as noted in Pinzón et al. (2022), fully satisfying the *Equalized Odds* criteria may trivialize the classifier i.e., a trivial classifier will make constant prediction rates regardless of the features of input samples.

The existence of the sharp decline in accuracy indicates that the FairFrontier may be non-convex, which is contradictory to a basic assumption in prior fair ML research that the loss function is convex. The proposed research can provide insights into the current evaluation paradigm and encourage the development of new benchmarking methods. Inferior model performance may arise from inadequate optimization which fails to attain the trade-off curve, or excessive optimization which leads the learning process to surpass local maximum fairness. The proposed accuracy upper bound will enable us to gauge whether fairness pursuit is inadequate or excessive and to evaluate the effectiveness of the performance trade-off.

## 5 BEYOND FAIRFRONTIER: ELIMINATING THE TRADE-OFF

Despite the dominant view of the tension between fairness and accuracy, a number of recent studies (Dutta et al., 2020; Langenberg et al., 2023) have shown that fairness and accuracy may benefit each other. In this section, we explore the possibility of going beyond FairFrontier and looking into possible conditions to achieve both complete fairness and maximal accuracy.

### 5.1 DECOMPOSING THE UNFAIRNESS

To start, we decompose unfairness into data unfairness and model unfairness (Dutta et al., 2020). We define data unfairness, stemming from inherent data disparities, as $F_{DU}$, and model unfairness originating from the design of the model architecture, as $F_{MU}$. They are formulated as follows (Dutta et al., 2020):

$$
\begin{aligned}
F_{DU} =& \frac{1}{2} \times |\text{TPR}^*_{A=0} - \text{TPR}^*_{A=1}| + \frac{1}{2} \times |\text{TNR}^*_{A=0} - \text{TNR}^*_{A=1}|, \\
F_{MU} =& \frac{1}{2} \times |((\text{TPR}_{A=0,T_\theta} - \text{TPR}^*_{A=0}) - (\text{TPR}_{A=1,T_\theta} - \text{TPR}^*_{A=1}))| \\
&+ \frac{1}{2} \times |((\text{TNR}_{A=0,T_\theta} - \text{TNR}^*_{A=0}) - (\text{TNR}_{A=1,T_\theta} - \text{TNR}^*_{A=1}))|.
\end{aligned}
\tag{12}
$$

where $T_\theta$ represents the choice of the classifier.

**Theorem 3** *The inequality $F_U \leq F_{DU} + F_{MU}$ holds true. Furthermore, if classifier $T_\theta$ is well-defined and satisfies one of these conditions where $S = \{T_{A=0}(x) \times T_{A=1}(x) \leq 0\}$:*

$$
\begin{aligned}
&\textit{Condition 1: } \forall x \in S, T^*_{A=0}(x) \geq 0, TPR^*_{A=0} < TPR^*_{A=1}, TNR^*_{A=0} > TNR^*_{A=1}. \\
&\textit{Condition 2: } \forall x \in S, T^*_{A=0}(x) \leq 0, TPR^*_{A=0} > TPR^*_{A=1}, TNR^*_{A=0} < TNR^*_{A=1}.
\end{aligned}
\tag{13}
$$

*then the equality $F_U = F_{DU} + F_{MU}$ holds with $F_{MU} > 0$.*

*Proof Sketch*: When one of these two conditions is satisfied, the expressions for both $F_{DU}$ and $F_{MU}$ maintain consistent sign conventions$(+/-)$ within their absolute value terms.

We also empirically prove Theorem 3. We conduct experiments on a synthetic dataset, whose generation process is outlined in Section 6.3. $X$ follows the Normal distribution, and the data across different groups is imbalanced. As shown in Figure 5, data unfairness (the grey curve) remains constant, in line with our definition 12 as it is independent of the design of the model architecture. Conversely, model unfairness (the green curve) fluctuates along with total unfairness (the red curve), signifying the dominance of this source of unfairness. It is worth noting that model unfairness may still exhibit the same variations in amplitude with total unfairness when the classifier resides outside the space $S$ defined in Definition 1, where $S = \{T_{A=0}(x) \times T_{A=1}(x) \leq 0\}$. This observation further underscores the effectiveness of our decomposition approach.

### 5.2 ELIMINATING THE ACCURACY-FAIRNESS TRADE-OFF

Theorem 3 suggests a potential solution to achieve complete fairness through a systematic debiasing approach that can address distinct sources of bias sequentially, rather than using a debiasing technique that focuses either on data or model unfairness. A similar approach has been advocated in prior research, e.g., Cheng et al. (2022). We first establish the following proposition and then propose a two-step streamlined approach to eliminate the accuracy-fairness trade-off.

**Proposition 1** *Given $F_{DU} = 0$, then the classifier for both complete fairness and maximal accuracy exists* iff. *the decision boundaries are identical across different sensitive groups, which is equivalent to $\forall x \in S, \mathbf{I}(T^*_{A=0}(x)) = \mathbf{I}(T^*_{A=1}(x))$.*

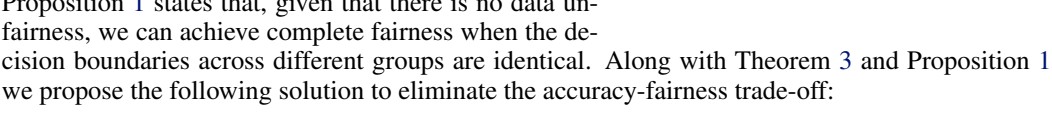

Figure 5: Unfairness decomposition where $\Delta$unfairness = Unfairness − model unfairness.

Proposition 1 states that, given that there is no data unfairness, we can achieve complete fairness when the decision boundaries across different groups are identical. Along with Theorem 3 and Proposition 1, we propose the following solution to eliminate the accuracy-fairness trade-off:

- **Step 1** Tackle data unfairness. Since data unfairness often stems from data imbalances, it can be rectified through data augmentation or sampling to achieve balanced data. For example, we can collect more data samples or features through active learning.

- **Step 2** Address model unfairness. Following Proposition 1, model unfairness can be addressed by transforming the data distribution to align decision boundaries across different groups.

Note that this proposition does not indicate that complete fairness is impossible under all circumstances with disparate decision boundaries, rather, it suggests that it is highly challenging to achieve complete fairness in these situations.

## 6 NUMERICAL EXAMPLES

In this section, we provide a series of results based on synthetic data to show the validity and feasibility of the proposed research. For simplicity, we posit the following assumption: Both the positive and negative prediction spaces are simply connected, i.e., any two samples with either positive labels or negative labels can be connected by a path. This assumption is reasonable because decision boundaries can be recognized as real-world standards, such as the criteria for conviction. Disconnected positive spaces imply disparate criteria for different people, suggesting potential discrimination.

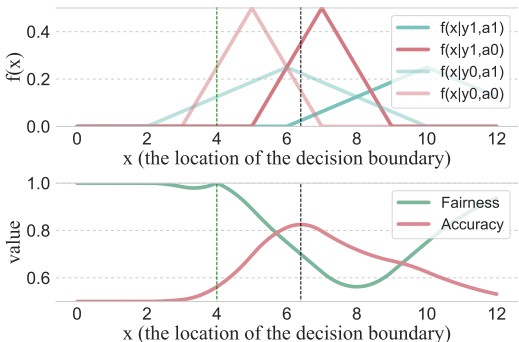

Figure 6: The figure above visualizes the data distributions in Example 3, where the optimal classifiers across different sensitive groups are **not identical**. The figure below shows the fairness and accuracy when the decision boundary changes. Besides, the green and the black dashed lines correspond to the optimal classifier for fairness and accuracy, respectively.

### 6.1 EXAMPLE 1. CONTINUOUS FAIRFRONTIER IN AN IDEAL SETTING

**Setting:** For the sensitive attribute $A = 1$, we set $X \mid A = 1, Y = 1 \sim \mathrm{N}(10, 2)$ and $X \mid A = 1, Y = 0 \sim \mathrm{N}(3, 2)$. Similarly, for the sensitive attribute $A = 0$ we set $X \mid A = 0, Y = 1 \sim \mathrm{N}(6, 2)$ and $X \mid A = 0, Y = 0 \sim \mathrm{N}(-1, 2)$. In addition, we have $\mathbb{P}(A = 1, Y = 1) = \frac{1}{2}, \mathbb{P}(A = 1, Y = 0) = \frac{1}{4}, \mathbb{P}(A = 0, Y = 1) = \frac{1}{8}$ and $\mathbb{P}(A = 0, Y = 0) = \frac{1}{8}$.

As depicted in Figure 2, we treat sensitive groups separately and obtain the FairFrontier. As fairness increases, the accuracy gradually declines at first. Beyond some extent of fairness, accuracy steeply declines yet maintains continuity. We believe this is due to the separate optimization of classifiers across different groups, which allows more flexibility to balance between complete fairness and maximal accuracy.

## 6.2 EXAMPLE 2. SHARP DECLINE IN ACCURACY

Under the same setting in Example 1, we demonstrate that over-pursuing fairness may contribute to a sharp decline in accuracy. As shown in Figure 4, fairness improves with little sacrifice of accuracy at first, but then the accuracy sharply declines by more than 0.2. After that, the trade-off curve exhibits an approximately linear decrement as fairness continues to rise. Besides, when a sharp decline in accuracy occurs, the classifier harms both sensitive groups, consistent with the empirical analysis in Hu & Chen (2020).

## 6.3 EXAMPLE 3. UNFAIRNESS DECOMPOSITION

**Setting:** For the sensitive attribute $A = 1$, we set $X \mid A = 1, Y = 1 \sim N(10, 3)$ and $X \mid A = 1, Y = 0 \sim N(2, 3)$. Similarly, for the sensitive attribute $A = 0$ we set $X \mid A = 0, Y = 1 \sim N(7, 3)$ and $X \mid A = 0, Y = 0 \sim N(-1, 3)$. In addition, we have $\mathbb{P}(A = 1, Y = 1) = \frac{1}{2}$, $\mathbb{P}(A = 1, Y = 0) = \frac{1}{4}$, $\mathbb{P}(A = 0, Y = 1) = \frac{1}{8}$ and $\mathbb{P}(A = 0, Y = 0) = \frac{1}{8}$.

As shown in Figure 5, we compute that $F_{DU} = 0.017$, and $F_U = F_{MU} + F_{DU}$. It is noted that in this example, unfairness is predominant by model unfairness, and $F_{MU}$ approximates $F_U$ when the decision boundary approaches the prediction space boundary. It is likely that both TPR and TNR come to parity across different groups when those two boundaries get close. Therefore, there exists a reversal in the sign of the terms within the absolute values in $F_{MU}$.

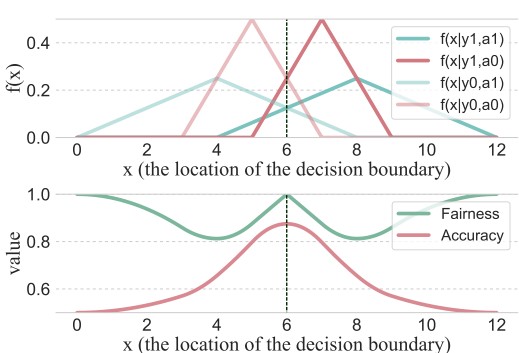

Figure 7: The figure visualizes the data distributions in Example 3, where the optimal classifiers across different sensitive groups are **identical**. The figure below shows the fairness and accuracy of different decision boundaries. The green and black dashed lines correspond to the optimal classifier for fairness and accuracy, respectively.

## 6.4 EXAMPLE 4. ELIMINATE THE ACCURACY-FAIRNESS TRADE-OFF

**Setting:** We denote the triangular distribution with lower limit $a$, upper limit $b$ and mode $c$ as the Triang$(a, b, c)$. For sensitive attribute $A = 1$, let $X \mid A = 1, Y = 1 \sim$ Triang$(4, 12, 8)$ and $X \mid A = 1, Y = 0 \sim$ Triang$(0, 8, 4)$. For the sensitive attribute $A = 0$ we set $X \mid A = 0, Y = 1 \sim$ Triang$(3, 7, 5)$ and $X \mid A = 0, Y = 0 \sim$ Triang$(5, 9, 7)$. Therefore, we can compute that the optimal classifiers for both groups are identical. Similarly, we set $X \mid A = 1, Y = 1 \sim$ Triang$(6, 14, 10)$, $X \mid A = 1, Y = 0 \sim$ Triang$(2, 10, 6)$, $X \mid A = 0, Y = 1 \sim$ Triang$(3, 7, 5)$ and $X \mid A = 0, Y = 0 \sim$ Triang$(5, 9, 7)$ for non-identical optimal classifiers for both sensitive groups. Besides, we choose $\mathbb{P}(A = 1, Y = 1) = \mathbb{P}(A = 1, Y = 0) = \mathbb{P}(A = 0, Y = 1) = \mathbb{P}(A = 0, Y = 0) = \frac{1}{4}$ for balanced datasets (Seen in Figures 6-7).

In this example, we successfully observe that when both TPR and TNR are equal across different groups, we can obtain both complete fairness and maximal accuracy *iff.* the optimal classifiers are identical(Shown in Figure 7). However, if the optimal classifiers across different groups are disparate, complete fairness may still hold (shown in Figure 6) while maximal accuracy becomes unattainable, with potentially poor prediction performance.

## 7  RELATED WORK

Research into the trade-off between fairness and accuracy has recently gained prominence, despite the substantial work in the field of fair machine learning (FairML)(Shui et al., 2022; Zhang et al., 2022b; Deng et al., 2022; Kang et al., 2022; Qi et al., 2022; Zhang et al., 2022a; Jiang et al., 2021; Liu et al., 2022; Zuo et al., 2022). Existing examinations of this trade-off can be classified into two main categories: data-centered approaches and distribution-centered approaches.

**Data-centered Approaches**. Progress in the field has witnessed substantial developments in data-centered approaches, with a strong emphasis on leveraging observational data. Kamiran & Calders (2011) first examined the nuanced interplay between accuracy and fairness, crafting an optimal classifier contingent upon the proportion of instances. Meanwhile, Chen et al. (2018) quantified unfairness via a comprehensive bias-variance decomposition, and Pinzón et al. (2022) delved into the geometric analysis of this trade-off with discrete data sources. However, these works fall short in characterizing the FairFrontier, and the specific shape of the FairFrontier remains unexplored.

**Distribution-centered Approaches.** These studies frequently assume that data distributions across different sensitive groups are known and accessible (Zhao & Gordon, 2019; Menon & Williamson, 2018; Blum & Stangl, 2019; Wang et al., 2023). Menon & Williamson (2018) derived the decision boundary by intuitively using the true positive rate of the selected classifier on sensitive attributes to measure unfairness. Blum & Stangl (2019) explored how fairness constraints on the training set can affect generalization performance when test set distributions differ. On the other hand, Dutta et al. (2020) assumed separate classifiers for different groups and derived the Chernoff bound to characterize the trade-off. However, these studies frequently focused more on the existence of the accuracy-fairness trade-off rather than the shape of the FairFrontier.

Recent works focusing on the trade-off often have distinct emphases tailored to specific scenarios. For instance, Zietlow et al. (2022) investigated the accuracy-fairness trade-off in computer vision, while Zhao (2021) studied it in fair regression. Besides, some works have explored the trade-off under scenarios involving distribution shift and optimization with privacy concerns(Wick et al., 2019; Pham et al., 2023; Lowy, 2023; Gultchin et al., 2022).

This work distinguishes itself from prior research by introducing rigorous theoretical analyses to delineate the FairFrontier and explore four potential curves that illustrate the trade-off between fairness and accuracy. For each category within the FairFrontier, we aim to delve into the underlying mechanisms and pinpoint the necessary conditions that give rise to these specific trade-offs. To achieve this goal, our approach centers on examining the positions of decision boundaries rather than focusing on classifiers, as previous studies have done. By characterizing the key properties of the FairFrontier, we gain a deeper understanding of the accuracy-fairness trade-off and ultimately aspire to eliminate this trade-off, thereby constructing an effective and fair ML system.

## 8  CONCLUSIONS AND FUTURE WORK

Since fairness has become an essential consideration in algorithmic decision-making, it is critical to discern the shape of the accuracy-fairness trade-off curve(FairFrontier). In this paper, we first investigated the ideal scenario where information in sensitive attributes can be fully captured by non-sensitive features and concluded that in most cases, the FairFrontier is continuous. We then further examined the shape of the FairFrontier in a more practical setting where non-sensitive attributes encode partial information in the sensitive attributes. We provided an upper bound to show that under certain conditions, accuracy may suffer a sharp decline when over-pursing fairness. Moreover, we went beyond the FairFrontier and decomposed the unfairness into data unfairness and model unfairness. A two-step streamlined approach was therefore proposed to eliminate the trade-off. Lastly, we provided several numerical examples to demonstrate our theoretical findings.

Looking forward, our proposed theoretical approach provides multiple avenues for future research. In this work, we assume the data distributions are known beforehand for analytical convenience, which is often impractical. Hence it is imperative to quantify unfairness decomposed through real-world datasets. In addition, the sharp decline in accuracy urges us to focus on the effectiveness of sacrificing performance for fairness and to carefully scrutinize the trade-off between our objectives and the well-being of the public.

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

# A  APPENDIX

In this section, we first obtain the Bayes optimal classifier for accuracy and fairness for binary classification. Then we prove the Lemmas, Theorems and the Proposition stated in the main manuscript.

## A.1  OPTIMAL CLASSIFIER FOR ACCURACY AND FAIRNESS

### A.1.1  OPTIMAL CLASSIFIER FOR ACCURACY

Following Menon & Williamson (2018) and Wang et al. (2023), we can write the Markov chain as $(A, Y) \to X \to \hat{Y}$. Then the following equation is obtained given the Bayes optimal classifier for accuracy:

$$
\begin{aligned}
Acc(T_\theta) =& \mathbb{P}(\hat{Y} = 1, Y = 1) + \mathbb{P}(\hat{Y} = 0, Y = 0) \\
=& \sum_a E_{x \sim f(x|Y=1,A=a)} \mathbf{I}(T_\theta(x)) \times \mathbb{P}(Y = 1, A = a) \\
&+ \sum_a E_{x \sim f(x|Y=0,A=a)} (1 - \mathbf{I}(T_\theta(x))) \times \mathbb{P}(Y = 0, A = a) \\
=& \sum_a E_{x \sim f(x)} \big( \frac{1}{f(x)} \times (f(x|Y=1, A=a) \times \mathbb{P}(Y = 1, A = a) \\
&- f(x|Y=0, A=a) \times \mathbb{P}(Y = 0, A = a)) \times \mathbf{I}(T_\theta)) + \mathbb{P}(Y = 0).
\end{aligned}
\tag{14}
$$

Hence, the optimal classifier for accuracy $T_\theta^*$ can be obtained by:

$$
\mathbf{I}(T_\theta^*) = \begin{cases} 1, & \sum_a f(x, Y = 1, A = a) \geq \sum_a f(x, Y = 0, A = a), \\ 0. & \sum_a f(x, Y = 1, A = a) < \sum_a f(x, Y = 0, A = a). \end{cases}
\tag{15}
$$

### A.1.2  OPTIMAL CLASSIFIER FOR FAIRNESS

Similar to the previous analysis, we can derive the optimal fairness-aware classifier with arbitrary weights $w$. Upon implementing the optimal classifier denoted as $T_\theta^f$ for classification, and considering the outcomes for each sensitive group $A = a$, where $\text{TPR}_{A=1} > \text{TPR}_{A=0}$ and $\text{TNR}_{A=1} > \text{TNR}_{A=0}$, we deduce the following:

$$
\begin{aligned}
F_U =& \omega_1 \times (\text{TPR}_{A=1} - \text{TPR}_{A=0}) + \omega_2 \times (\text{TNR}_{A=1} - \text{TNR}_{A=0}) \\
=& \omega_1 \times (E_{x \sim f(x|Y=1,A=1)} \mathbf{I}(T_\theta(x)) - E_{x \sim f(x|Y=1,A=0)} \mathbf{I}(T_\theta(x))) \\
&+ \omega_2 \times (E_{x \sim f(x|Y=1,A=1)} (1 - \mathbf{I}(T_\theta(x)))) - E_{x \sim f(x|Y=1,A=0)} (1 - \mathbf{I}(T_\theta(x))) \\
=& E_{x \sim f(x)} (\lambda_1 - \lambda_2) \times \mathbf{I}(T_\theta)).
\end{aligned}
\tag{16}
$$

where

$$
\begin{aligned}
\lambda_1 = \omega_1 \times (f(x|Y=1, A=1) - f(x|Y=1, A=0)), \\
\lambda_2 = \omega_2 \times (f(x|Y=0, A=1) - f(x|Y=0, A=0)).
\end{aligned}
\tag{17}
$$

Hence, the optimal classifier for accuracy $T_f^*$ can be achieved by:

$$
\mathbf{I}(T_\theta^f) = \begin{cases} 1, & \lambda_1 \geq \lambda_2, \\ 0. & \lambda_1 < \lambda_2. \end{cases}
\tag{18}
$$

Alternatively, if $\text{TPR}_{A=1} > \text{TPR}_{A=0}$ and $\text{TNR}_{A=1} < \text{TNR}_{A=0}$, we obtain that:

$$
\mathbf{I}(T_\theta^f) = \begin{cases} 1, & \lambda_1 \geq -\lambda_2, \\ 0. & \lambda_1 < -\lambda_2. \end{cases}
\tag{19}
$$

## A.2  PROOFS

### A.2.1  PROOF OF LEMMA 1

*proof.* The proof for the case of the sharp decline in both fairness and accuracy (the brown curve in Figure 8) is analogous to the case of the sharp decline in fairness(the grey curve in Figure 8).

Therefore, we only prove that the sharp decline in fairness cannot occur, regardless of whether sensitive attributes are encoded or not.

As is depicted in Figure 9, Let point $M$ represent the location on the trade-off curve where accuracy is maximized. Additionally, point $B$ denotes the point discontinuity where the curve is left-continuous but right-discontinuous, and point $A$ represents the point discontinuity where the curve is right-continuous but left-discontinuous.

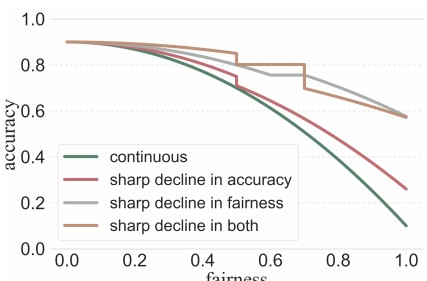

Now we prove that the accuracy at point A must be the maximum accuracy. If not, there must exist $S \subseteq \Omega$, $\forall x \in S, T_A(x) \times T_M(x) < 0$. Consequently, we can formulate the classifier $T'_a$ as follows:

$$T'_A(x) = \begin{cases} T_A(x), & x \in \Omega - S, \\ T_M(x). & x \in S. \end{cases} \quad (20)$$

where $\Omega$ represents the probability space. Therefore $Acc(T'_A) > Acc(T_A)$ This result conflicts with which the trade-off curve at the point discontinuity is left-discontinuous. Therefore, a sharp decline in fairness cannot occur, regardless of whether sensitive attributes are encoded or not.

Figure 8: Four shapes of FairFrontier. "Green" delineates a continuous frontier, "Red" exhibits a sharp decline in accuracy when over-pursuing fairness, and "Grey" shows a sharp decline in fairness when improving accuracy. "Brown" represents a sharp decline in both accuracy and fairness.

The lemma can also be derived from the properties of envelope curves: According to the characteristics of envelope curves, it is known that the Pareto Frontier is formed by the envelope of a cluster of possible accuracy-fairness curves. If the envelope curve exhibits a discontinuity at point A, then any accuracy-fairness curve passing through point A within the cluster will also be discontinuous.

### A.2.2 PROOF OF LEMMA 2

*Full Description.* If the sharp decline in accuracy occurs (the red curve in Figure 8), *iff.* the right-discontinuous point satisfies:

(1) Condition 1:Given that classifier $T_A$ operating on different sensitive groups, we have:

$(\text{TPR}_{A=1} - \text{TPR}_{A=0})(\text{TNR}_{A=1} - \text{TNR}_{A=0}) \geq 0.$

(2) Condition 2:$Acc(T_A) = \max Acc(T_\theta)$, s.t. $F_U(T_\theta) = F_U(T_A)$.

(3) Condition 3:

if $\text{TPR}_{A=1} \geq \text{TPR}_{A=0}$, $\text{TPR}_{A=1} \geq \text{TPR}_{A=0}$,

then

$$\hat{Y} = \begin{cases} \mathbf{I}(f(x|y{=}1,a{=}1) \leq f(x|y{=}0,a{=}1)), & A = 1, \\ \mathbf{I}(f(x|y{=}1,a{=}0) \geq f(x|y{=}0,a{=}0)), & A = 0. \end{cases}$$

else if $\text{TPR}_{A=1} \leq \text{TPR}_{A=0}$, $\text{TPR}_{A=1} \leq \text{TPR}_{A=0}$,

then

$$\hat{Y} = \begin{cases} \mathbf{I}(f(x|y{=}1,a{=}1) \geq f(x|y{=}0,a{=}1)), & A = 1, \\ \mathbf{I}(f(x|y{=}1,a{=}0) \leq f(x|y{=}0,a{=}0)), & A = 0. \end{cases}$$

$$(21)$$

where $\mathbf{I}$ is the characteristic function which states that if $x \geq 0$ is true, $\mathbf{I}(x) = 1$; else, $\mathbf{I}(x) = 0$.

*proof.* We initially establish the necessity of the conditions and subsequently prove their sufficiency.

As is depicted in Figure 10, Let point $M_A$ represent the location on the trade-off curve where accuracy is maximized. Additionally, point $B$ denotes the point discontinuity where the curve is right-continuous but left-discontinuous, and the curve is left-continuous but right-discontinuous.

**Step 1:** Proof for Condition 1

For condition 1, we hypothesize that it is not satisfied. We denote $T_a$ as the classifier operating on the data distribution of group $A = a$ at the point discontinuity. Without loss of generality, we

assume that: $\text{TPR}_{A=1} > \text{TPR}_{A=0}, \text{TNR}_{A=1} < \text{TNR}_{A=0}$, then we have $\{T_{A,A=1}(x) < 0\} \neq \Omega, \{T_{A,A=0}(x) > 0\} \neq \Omega$ where $\Omega$ represents the probability space. Otherwise, $0 = \text{TPR}_{A=1} > \text{TPR}_{A=0} \geq 0, 0 = \text{TNR}_{A=0} \geq \text{TNR}_{A=1} \geq 0$, which cannot hold. Consequently, there exists $S \subseteq \Omega, \forall x \in S, T_A(x) \times T_M(x) < 0$ and we can formulate the classifier $T_a'$ as follows:

$$T_A'(x) = \begin{cases} T_A(x), & x \in \Omega - S, \\ -T_A(x). & x \in S. \end{cases} \tag{22}$$

where $\Omega$ represents the probability space. Therefore $F_U(T_A) > F_U(T_A')$, which is right-continuous on the FairFrontier.

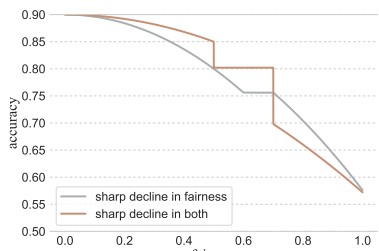

**Step 2:** Proof for Condition 2. This condition must be satisfied; otherwise, point A would not lie on FairFrontier.

**Step 3:** Proof for Condition 3. This condition can be obtained through a similar analysis to A.1.2.

**Step 4:** Proof for Sufficiency. Condition 2 indicates that the point discontinuity lies on the trade-off curve. According to Step 3, we conclude that fairness at the point discontinuity attains the local maximum, where $F_U(T_\theta^f) = F_U(T_A)$. Considering that the accuracy at the point discontinuity is typically unique, we obtain that $T_A = T_\theta^f$. Then point discontinuity represents the point discontinuity where the curve is left-continuous but right-discontinuous. Now we can say that there exists a sharp decline in accuracy.

Figure 9: The grey curve represents the sharp decline in fairness when over-pursuing accuracy and the brown curve delineates the sharp decline in both accuracy and fairness.

*Notes.* In most cases, Condition 1 and Condition 3 cannot be simultaneously satisfied. For instance, let's define group $A = 0$ as the unprivileged group, whose true positive prediction rate(TPR) and true negative prediction rate(TNR) are both lower than group $A = 1$. If both conditions are met, it means that the performance of the perfect prediction for the unprivileged group would be worse than that of the adversely worst prediction for the privileged group, which is seldom achieved.

### A.2.3  PROOF OF THEOREM 1

*proof.*  If $T_\theta$ is the optimal classifier for both accuracy and fairness, the boundary condition for maximum accuracy is achieved. If complete fairness is attainable, then we obtain that $F_U = 0$ which corresponds to Condition 1. Besides, given that the boundary condition for maximum fairness is achieved, we obtain Condition 2 through the result in A.1.2. Therefore, the general conditions are satisfied if the classifier $T_\theta$ maximizes both accuracy and fairness simultaneously.

Given that the distributions across different groups are balanced, i.e. $\mathbb{P}(a,y) = \frac{1}{4}$, for any sample $x$ on the decision boundary $B$, we have:

$$\begin{aligned} &\sum_a f(x \mid Y = 1, A = a) \times \mathbb{P}(Y = 1, A = a) \\ =&\sum_a f(x \mid Y = 1, A = a) \times \frac{1}{4} \\ =&\sum_a f(x \mid Y = 0, A = 0) \times \mathbb{P}(Y = 0, A = a) \\ =&\sum_a f(x \mid Y = 0, A = 0) \times \frac{1}{4} \end{aligned} \tag{23}$$

Therefore, we can readily derive the corresponding conditions by incorporating Condition 2 with the previously obtained result.

### A.2.4  PROOF OF THEOREM 2

*proof.*  We prove this result in three steps:

**Step 1**: Classifiers that share the same extent of fairness characteristics cannot traverse the enclosed space $S$. The proof can be seen below:

There exists $S \subseteq \Omega, \forall x \in S, T_A(x) \times T_M(x) < 0$ and we can formulate the classifier $T'_a$ as follows:

$$T'_A(x) = \begin{cases} T_A(x), & x \in \Omega - S, \\ -T_A(x). & x \in S. \end{cases} \tag{24}$$

where $\Omega$ represents the probability space. Therefore $F_U(T_A) < F_U(T'_A)$ and and $Acc(T_A) < Acc(T'_A)$. Hence, there always exists a classifier $T'_\theta$ with little adjustment that dominates $T^f_\theta$.

**Step 2**: Classifiers with the same extent of fairness cannot reside within the space $S$. Given that the optimal classifier for fairness $T^f_\theta$, obtained through A.1.2, is well-defined. Besides, for any well-defined classifier $T_\theta$, it must reside within the space $S$ where $S = \{T_{A=0}(x) \times T_{A=1}(x) \leq 0\}$. Hence, the classifier with the same extent of fairness cannot reside within the space $S$.

**Step 3**: The classifier with the same extent of fairness can only exist outside the space $S$, but its accuracy must be inferior to that of the optimal classifier for each sensitive group. Because the classifier outside the space $S$ always yields worse performance compared to well-defined classifiers, including the optimal classifiers for both sensitive groups. Consequently, the original inequality remains valid, as does the derived one.

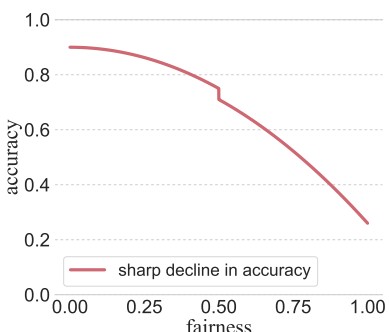

### A.2.5 PROOF OF THEOREM 3

*proof.* We begin by proving the inequality, and subsequently, we establish the equality by leveraging the well-defined classifier. It stipulates that beyond the region enclosed by the decision boundaries of the optimal classifiers for distinct sensitive groups, denoted as $S = T_{A=0}(x) \times T_{A=1}(x) \leq 0$, the sign $(+/-)$ of the predictions aligns with the optimal classifier employed for each respective group.

Figure 10: The red curve represents the sharp decline in accuracy when overpursuing fairness.

$$\begin{aligned} F_U =& \frac{1}{2} \times |\text{TPR}_{A=0,T_\theta} - \text{TPR}_{A=1,T_\theta}| + \frac{1}{2} \times |\text{TNR}_{A=0,T_\theta} - \text{TNR}_{A=1,T_\theta}| \\ =& \frac{1}{2} \times |(\text{TPR}_{A=0,T_\theta} - \text{TPR}^*_{A=0}) - (\text{TPR}_{A=1,T_\theta} - \text{TPR}^*_{A=1}) + (\text{TPR}^*_{A=0} - \text{TPR}^*_{A=1})| \\ &+ \frac{1}{2} \times |(\text{TNR}_{A=0,T_\theta} - \text{TNR}^*_{A=0}) - (\text{TNR}_{A=1,T_\theta} - \text{TNR}^*_{A=1}) + (\text{TNR}^*_{A=0} - \text{TNR}^*_{A=1})| \\ \leq& F_{MU} + F_{DU}. \end{aligned} \tag{25}$$

where the last inequality can be obtained by applying the absolute value inequality.

In addition, since $T_\theta$ is well-defined, we can relax the absolute value. Specifically, if $T^*_{A=0}(x) > 0$ when $x \in S$, then:

$$\begin{aligned} \text{TPR}^*_{A=0} - \text{TPR}_{A=0,T_\theta} > 0, \quad & \text{TPR}^*_{A=1} - \text{TPR}_{A=1,T_\theta} < 0, \\ \text{TNR}^*_{A=0} - \text{TNR}_{A=0,T_\theta} < 0, \quad & \text{TNR}^*_{A=0} - \text{TNR}_{A=0,T_\theta} > 0. \end{aligned} \tag{26}$$

else if $T^*_{A=0}(x) < 0$ when $x \in S$, then:

$$\begin{aligned} \text{TPR}^*_{A=0} - \text{TPR}_{A=0,T_\theta} < 0, \quad & \text{TPR}^*_{A=1} - \text{TPR}_{A=1,T_\theta} > 0, \\ \text{TNR}^*_{A=0} - \text{TNR}_{A=0,T_\theta} > 0, \quad & \text{TNR}^*_{A=0} - \text{TNR}_{A=0,T_\theta} < 0. \end{aligned} \tag{27}$$

We can obtain that $F_{MU} > 0$. Besides, when one of the conditions of this theorem is satisfied, both $F_{DU}$ and $F_{MU}$ will have the same sign within the absolute value, regardless of whether it is for TPR or TNR. Hence, we can conclude that: $F_U = F_{DU} + F_{MU}$.

### A.2.6 PROOF OF PROPOSITION 1

*proof.* We initially establish the sufficiency of the conditions and subsequently prove their necessity.

**Sufficiency:** Given $\forall \in S, \mathbf{I}(T^*_{A=0}(x)) = \mathbf{I}(T^*_{A=1}(x))$, we let $T_\theta = T^*_{A=0} = T^*_{A=1}$. It can be easily proved that the classifier is well-defined and the enclosed space $S = \emptyset$ where $S = \{T_{A=0}(x) \times T_{A=1}(x) \leq 0\}$. Therefore, $F_{MU} = 0$. Given $F_{DU} = 0$, we can obtain that $F_U = 0$. Since this chosen classifier is optimal for each group, we have:

$$
\begin{aligned}
Acc(T_\theta) =& Acc_{A=0}(T_\theta) \times \mathbb{P}(A = 0) + Acc_{A=1}(T_\theta) \times \mathbb{P}(A = 1) \\
=& Acc_{A=0}(T^*_{A=0}) \times \mathbb{P}(A = 0) + Acc_{A=1}(T^*_{A=1}) \times \mathbb{P}(A = 1).
\end{aligned}
\tag{28}
$$

where $Acc_a(T_\theta)$ is denoted as the accuracy of the group $A = a$ obtained by the classifier $T_\theta$. In light of Theorem 4 in (Lipton et al., 2018), we can obtain that the maximum accuracy is achieved under the optimal classifier for both sensitive groups.

**Necessity:** Once maximum accuracy has been attained, the classifier becomes the optimal classifier, where $T_\theta = T^*_\theta$, thus affirming the chosen classifier's well-defined nature. Therefore, the equation $F_U = F_{MU} + F_{DU} = 0$ holds true and we can obtain that $F_{MU} = 0$. According to Theorem 1, when $S \neq \emptyset$ which means that the optimal classifier for different groups is not identical, we can obtain the inequality $F_{MU} > 0$, which is inconsistent with our previous result. We then conclude that the optimal classifier for different groups must be identical.

