# OpenReview forum: "A Theoretical Approach to Characterize the Accuracy-Fairness Trade-off Pareto Frontier"
_ICLR.cc/2024/Conference — Submitted to ICLR 2024_

### Official Review · Reviewer_HvQJ · 2023-10-31

**Soundness:** 2 fair
**Presentation:** 2 fair
**Contribution:** 2 fair
**Rating:** 3
**Confidence:** 3

**Summary:**

This paper provided theoretical analyses for the accuracy-fairness trade-off and characterized the shape of the accuracy-fairness trade-off Pareto frontier (termed FairFrontier) for fair machine learning problems. The authors made an attempt to address a fundamental question “For a given data distribution, what would the accuracy-fairness trade-off curve look like?”. They leveraged synthetic data from known data distributions and constructed optimal classifiers to conduct examinations of the continuity of Pareto frontiers. Specifically, they characterized four different shapes of FairFrontier in terms of continuity and analyzed under what conditions these different frontiers will occur.

**Strengths:**

The paper does introduce some new perspectives to the fair machine learning community. The main problems the authors attempted to address include:
- Two of the four Pareto front curves cannot occur, i.e., the sharp decline in fairness and the
sharp decline in both accuracy and fairness. (Lemma 1)
- When does a sharp decline in accuracy occur? (Lemma 2)
- When it is possible to maximize both accuracy and fairness simultaneously? (Theorem 1, Proposition 1)
- When do we expect to see a sharp decline in accuracy? (Theorem 2)
- When model unfairness and data unfairness are completely decoupled and additive? (Theorem 3).

**Weaknesses:**

However, this paper should be rejected because (1) the theoretical analysis has limited novelty. (2) the theories developed in this paper are not solid and convincing. (3) the paper is imprecise and unpolished. (4) the insight to the real trade-off construction from the presented theories is limited.

**Questions:**

**Main arguments**

1. The shape of Pareto fronts between accuracy and fairness is largely dependent on the model architectures and fairness metrics. Many research could be found in the literature that focuses on capturing the trade-off curve for different ML models. Ignoring imprecise presentation of Theorem 1-2, Theorem 1, and Proposition 1 convey a trivial argument that if optimizers for different sensitive groups are aligned, there is no conflict between fairness and accuracy. I cannot see enough novelty from there.
2. Theorem 2: if $T_\Theta^f$ is the optimal classifier for fairness, it corresponds to an extreme point on the global Pareto front with the lowest accuracy. I don’t see the possibility of the case where accuracy of  $T^*_a$ is lower than that of the optimal classifier for fairness.
3. The statement of eliminating trade-offs is not rigorous. Also the authors didn’t give an actionable solution for aligning decision boundaries across different groups. Since the same accuracy classifier means the same decision boundary, can we just use the same classifier for different groups? In Proposition 1, the author basically formalized or rephrased when the accuracy-fairness conflict doesn’t exist.  When the decision boundaries are identical across different groups, it means the sensitive attribute distributions are unbiased across positive and negative groups. If the distributions across different groups are biased, is it still possible to align decision boundaries with a separate classifier for each group? It is highly doubted.
4. Section 6.1 and 6.2: using the same setting, it is not clear to the readers how the two different shapes of the Pareto front in Figure 2 and Figure 4 were obtained.
5. Section 6.4: Example 4 in Figure 6, where is the claim “maximum accuracy becomes unattainable” based on? Again, the elimination of the accuracy-fairness trade-off doesn’t come from any changes in model architectures but comes from the natural optimization over accuracy. The example in Figure 7 seeing the same fairness and accuracy optimizer is due to the unbiased distribution of sensitive attributes across positive and negative groups.

**The paper has many imprecise parts, here are a few:**

1. Theorem 1 equation (7): Based on my understanding, it means the two groups are not discriminative at all.
2. Theorem 1 Condition 2: arithmetic operation over probability distribution function f(**) makes no sense.* f(*) is more like a mean function of nonsensitive feature vector, which limits the generality of the derived theory to any data distribution.
3. Theorem 1 P5 Condition 1: the equality between the 2nd and 3rd terms means basically the sensitive feature is completely aligned with target labels? In addition to (7), the theorem tried to convey that we can find a unique optimizer for both accuracy and fairness with completely random data samples from independent uniform distributions (for A and Y). I don’t see much useful insight from this theorem.
4. P6 second paragraph “a basic assumption in prior fair ML research that the loss function is convex”: the non-convexity of the Pareto front between accuracy and fairness doesn’t necessarily mean non-convexity of loss functions. The loss function is an estimation of accuracy and could be convex or non-convex depending on the ML models, while accuracy is a discrete measure.
5. P6 equation (12) what is the definition of $TPR^*$  and how it is computed are not clear.
6. Section 5 P7 the claim about how to tackle data unfairness doesn’t completely capture the root causes of data fairness. Other than data imbalance, there is inherent data bias from biased ground truth labels. There are a bunch of pre-processing based fair ML methods where the ground truth labels were re-labeled to reduce data bias.

**Things to improve the paper that did not impact the score:**

1. section 2 on p2 Notation: $\Theta$ in $T_{\Theta}$  is not introduced before use.
2. section 2 on p3 “FairFrontier is monotonically non-increasing” is misleading when describing a curve of two dimensions. From the point of view of accuracy, it is non-increasing, while in terms of fairness, it is non-decreasing. And the monotonicity is self-explained by the definition of Pareto fronts.
3. Figure 4 y axis label needs to be fixed.
4. P5 Section 4.2 second paragraph: “…the sign of the predictions is the same as that of the optimal classifier used for each group” → the optimal classifier for the whole dataset?
5. Figures 6-7 captions are misleading: both figures are demonstration for Example 4.

---

### Official Review · Reviewer_vpks · 2023-11-01

**Soundness:** 1 poor
**Presentation:** 2 fair
**Contribution:** 2 fair
**Rating:** 3
**Confidence:** 4

**Summary:**

This paper analyzes the accuracy-fairness tradeoff (induced by Pareto efficient classifiers) in a binary classification setting. Specifically, the paper analyzes the continuity of this curve in different scenarios when the non-sensitive attributes fully (vs. partially) capture the sensitive attributes. The paper further proposes a methodology that provides guidelines on how to arrive in settings in practice where the maximum accuracy coincides with maximum fairness (i.e., eliminating the tradeoff). Simulation studies corroborate the theoretical results.

**Strengths:**

1. Understanding the accuracy-fairness tradeoff is an important and interesting problem.

2. The paper establishes theoretical results that identify different scenarios that lead to different continuity behaviors of this tradeoff curve, providing new insights towards understanding this tradeoff curve.

**Weaknesses:**

1. Clarity: The writing and presentation of the paper can be improved. I struggle to understand the setup and the main results of the paper.

a. Definitions:

(i) What do "overall distribution" and "classifier for the overall distribution" mean? What does "classifier for one group" mean? (Does it mean this classifier cannot be applied to the other group)?

(ii) "Notation" of Section 2 defines \hat{Y} based on the value of of T_a(x). What happens if T_a(x) = 0?

(iii) Section 2 states that "other definitions of fairness can also be incorporated into our analysis". This needs to be more concrete. Which results hold under other definitions, and what are the requirements on these fairness definitions?

(iv) Where are TPR* and TNR* defined in Eq. 12?

b. The theoretical results are often stated in loose terms. While I appreciate the intuitive language, I struggle to understand what they mean mathematically.

(i) In Lemma 1, what do "sharp decline" and "sensitive attributes are encoded" mean mathematically?

(ii) In Lemma 2, what do "over-pursuing fairness", "local maximum of fairness", and "highly unfavorable prediction for one group" mean mathematically? What does "mostly continuous" mean mathematically?

(iii) In Theorem 1, does "one of the following" mean "at least one of the following" for Eq. 8? Does "we obtain one of the following conditions" mean "it satisfies at least one of the following conditions" for Eq. 9?

(iv) In Theorem 2, what does it mean for a classifier to be "non-trivial"? What does it mean for the sign relationship to be consistent (what happens when equal to 0)?

(v) In Proposition, what does the indicator function mean (what happens when equal to 0)?

===

2. Proposed approach: The proposed approach is very abstract and conceptual, and I struggle to see the intuition behind it. For example, why should one tackle data unfairness first followed by model unfairness? How to "transform" the data in Step 2?

===

3. Experiments: More explanation on how the experiments match the theoretical formulation would be helpful. For example:

(i) why do the distributions used in Sec 6.1 correspond to the ideal setting?

(ii) What does "value" mean in the figures?

(iii) Why are triangular distributions used in Sec 6.4?

===

4. Minor:

- P2: "Non-sensitive" -> "non-sensitive"
- P2: Need to specify the constraint that w1 + w2 = 1.
- Eq 4: Need to write out "y=1" as opposed to just "y".
- P6: "empirically prove Theorem 3" -> "empirically validate"

**Questions:**

I'm quite confused about the setup and main results of the paper. Clarifications on Weaknesses 1 and 3 are helpful.

**Details Of Ethics Concerns:**

I appreciate that the paper studies a fairness problem. It would be helpful to discuss the limitations of the proposed two-step approach. For example, is it possible that this approach introduces new unfairness to one subpopulation? What are possible alternative approaches? (The proposed two-step approach is very abstract so it might not introduce much harm, but at the same time, I wonder how useful it can be.)

---

### Official Review · Reviewer_aY8j · 2023-11-01

**Soundness:** 2 fair
**Presentation:** 1 poor
**Contribution:** 3 good
**Rating:** 6
**Confidence:** 4

**Summary:**

Accuracy-fairness tradeoffs have been of considerable interest. This work proposes a theoretical framework by further characterizing the *shape* of the accuracy-fairness trade-off Pareto frontier. They call this the FairFrontier, i.e., the set of all optimal Pareto classifiers that no other classifiers can dominate. Of particular importance are two main findings:
(i) When the sensitive attributes can be fully interpreted by non-sensitive attributes, the Pareto frontier is mostly continuous.
(ii) Accuracy can have a sharp decline.
They also include experiments on synthetic datasets demonstrating these findings.

**Strengths:**

This work addresses an interesting problem of determining/understanding the shape of the accuracy-fairness frontier for equalized odds. This is an important step in the research on fairness-accuracy tradeoffs.
The proof techniques have interesting ideas from envelope curves which is quite novel.
They focus on an important aspect: the continuity of the Pareto frontier which has not been looked at before.

**Weaknesses:**

Lemma 1 and 2 are not written very rigorously. When they say sharp decline, it can mean different things. For instance, it can be thought of as a sharp change in the partial derivative while still being continuous. In this case, the authors specifically mean discontinuity? I think this should be written explicitly in the Lemma. While I understand that they are trying to emphasize its significance in the fairness context, it is still important to be nonambiguous about what is meant.

On the other hand, the Theorem statements are the opposite - they seem to be stating the conditions without explaining what they mean or would imply.

They talk about envelope curves but no references are discussed.

FairFrontier may be non-convex seems like too strong of a claim. Is there enough evidence for this? I would recommend including only claims that are substantiated by evidence. I am just wondering if there might even be a proof for convexity. For instance, consider two classifiers C1 and C2 on the Pareto front. Now consider a new classifier that chooses C1 w.p. a and C2 w.p. 1-a. Shouldn't this always lie within the Pareto front?

"overpursue fairness" is quite a confusing term

The paper would also benefit from a nice toy example that explains the intuition of the results early on. The experimental section is not well described with pointers to figures that have already appeared before. The figure captions and axes do not have enough details.

There are no experiments on real datasets.

**Questions:**

Clarification on non-convexity.

Clarification of Lemmas - with nonambiguous statements.

Clarification of Theorems with more concise statements.

Experiemental section is quite confusing to follow. More discussion and intuition on what to expect is necessary.

---

### Official Review · Reviewer_Ypkg · 2023-11-01

**Soundness:** 1 poor
**Presentation:** 2 fair
**Contribution:** 1 poor
**Rating:** 3
**Confidence:** 4

**Summary:**

The paper attempts to provide a theoretical characterization of the fairness and accuracy trade-off in a binary classification setting.  The authors claim the following contributions. 1) When non-sensitive features fully encode the sensitive features, the paper proves that the Pareto frontier graph is continuous in the fairness parameter. 2) When non-sensitive features do not fully encode sensitive features, the Pareto frontier may be discontinuous in the accuracy parameter. and 3) Authors claim that by decomposing the unfairness in data and model unfairness one can eliminate the fairness and accuracy trade-off.

**Strengths:**

The paper attempts to solve the relevant and important problem. The approach that looks at the fairness/accuracy trade-off in the light of how closely non-sensitive attributes encode the sensitive attributes has some potential and is worth pursuing in my opinion. Having said that, the paper lacks rigour and clarity. See the weaknesses for detailed comments.

**Weaknesses:**

The paper can be improved significantly in both the technical content and writing.

1. The authors claim in the abstract that they demonstrate that the trade-off exists in real-world settings. However, they do not consider any real-world example or experiment in the paper.
2. Many terms in the paper are used without rigorously defining them. For instance, what do you mean by over-pursuing fairness? what are mostly continuous functions?
3. Definition 1 says that the well-defined classifier must agree with optimal classifiers for both groups on the set over which they agree with each other; even if they are inaccurate. I do not think this is a practical assumption.
4.  Many claims in the paper are not substantiated. For instance, the authors say in the first line of Section 3 that the Pareto frontier is convex without proving or referring to any work in literature. They contradict themselves in the adjoining figure by plotting a concave Pareto frontier.
5. The proofs in the paper are handwavy at best. For instance, the proof of Lemma 1 is sloppy. Though I am taking the example of Lemma 1,  other proofs also lack significantly in rigour.
- No points  M, A and B in Figure 9.
- Why define B if you are not using it in the proof?
- What do you mean by sharp decline? A rigorous mathematical term should be used in formal proofs.
- I am quite certain that $\Omega$ is not a probability space. If it is, what is the outcome space, probability measure and signa-algebra that defines this space? Also, it is not at all clear why $x$ lies in this space.  Isn't $\Omega$ a feature space?
6. Theorem 1 seems a straightforward consequence of the result by Alexandra Chouldechova from the paper titled " Fair prediction with disparate impact: A study of bias in recidivism prediction instruments" (also see, Inherent Trade-Offs in the Fair Determination of Risk Scores, Kleinberg et.al)

**Questions:**

See weaknesses.

---

### Meta-Review · Area_Chair_FytG · 2023-12-10

**Metareview:**

The paper aims to theoretically characterize the trade-off between fairness and performance in binary classification. I agree with the reviewers that the general motivation of the paper is interesting but the paper is definitely not ready for publication. Thus, I encourage the authors to revisit their paper and implement the feedback provided by the reviewers to improve it before resubmitting.

**Justification For Why Not Higher Score:**

Clear rejection case, with not rebuttal.

**Justification For Why Not Lower Score:**

N/A

---

### Decision · Program_Chairs · 2024-01-16

Reject